# Retinal imaging with hand-held optical coherence tomography in older people with or without postoperative delirium after hip fracture surgery: A feasibility study

Abiodun M. Noah [1]*, Jennie Spendlove[1], Zhanhan Tu[2], Frank Proudlock[2], Cris S. Constantinescu[3,4], Irene Gottlob[2,4], Dorothee P. Auer[5,6], Rob A. Dineen [5,6], Iain K. Moppett[1]

1 Anaesthesia and Critical Care, Injury, Recovery and Inflammation Sciences, University of Nottingham, Nottingham, United Kingdom, 2 Ulverscroft Eye Unit, Department of Neuroscience, Psychology and Behaviour, University of Leicester, Leicester, United Kingdom, 3 Clinical Neurology, Mental Health and Clinical Neurosciences, University of Nottingham, Nottingham, United Kingdom, 4 Cooper Neurological Institute, Cooper Medical School of Rowan University, Camden, NJ, United States of America, 5 Radiological Sciences, Mental Health and Clinical Neurosciences, University of Nottingham, Nottingham, United Kingdom, 6 Nottingham NIHR BRC, School of Medicine, University of Nottingham, Nottingham, United Kingdom

* Abi.noah@nottingham.ac.uk

## Abstract

### Introduction

Postoperative delirium in older people may result from the interaction between intrinsic brain vulnerability (e.g. neurodegeneration) and precipitating factors (e.g. surgery induced cytokines). Intrinsic brain vulnerability may be overt (e.g. Alzheimer's disease) or preclinical. In cognitively intact older people presenting for surgery, identification of preclinical neurodegeneration using bedside tools could aid postoperative delirium risk stratification. Thinning of the circumpapillary retinal nerve fibre layer thickness is associated with neurodegenerative disorders e.g. Alzheimer's disease. We propose that thinning of the retinal nerve fibre layer may be present some older people with postoperative delirium due to preclinical neurodegeneration, albeit to a lesser extent than in overt dementia.

### Objectives

The primary objective: Feasibility of acquiring usable retinal images with the hand-held optical coherence device, at the bedside of older, hip fracture surgery patients. Secondary objective: Comparison of the circumpapillary retinal nerve fibre layer thickness between people who did/did not have postoperative delirium. Proportion of exclusions due to retinal pathology.

### Method

Feasibility study involving 30, cognitively intact, older people recovering from hip fracture surgery. Retinal images were obtained using the hand-held optical coherence tomography

**Data Availability Statement:** All relevant data are within the paper and its Supporting information.

**Funding:** This study was funded by a British Journal of Anaesthesia/Royal College of Anaesthetists project grant (WKR0-2015-0066). The grant recipient is IKM. Supplementary funding to support purchase of the Handheld Optical coherence tomography device was provided by the Precision Imaging Beacon Centre, University of Nottingham and Academic Department of Neurology, University of Nottingham.

**Competing interests:** The authors have declared that no competing interests exist.

device at the participants' bedside. Imaging was deferred in participants who had postoperative delirium.

## Results

Retinal images that could be assessed for circumpapillary retinal nerve fibre layer thickness were obtained in 26 participants (22 no postoperative delirium, 4 postoperative delirium). The mean circumpapillary retinal nerve fibre layer thickness was lower in the participants who had postoperative delirium compared to those who did not experience postoperative delirium (Mean (95% CI) of 76.50 (62.60–90.40) vs 89.19 (85.41–92.97) respectively).

## Conclusion

Retinal imaging at the patient's bedside, using hand-held OCT is feasible. Our data suggests that the circumpapillary retinal nerve fibre layer may be thinner in older people who experience postoperative delirium compared to those who do not. Further studies are required.

## Introduction

### Background

Postoperative delirium (POD), the most common neuropsychiatric complication of surgery in older people, i.e. age $\geq$ 65 years, occurs in up to 53% of patients recovering from hip fracture surgery and has devastating consequences [1]. POD is an independent predictor of subsequent new diagnosis of dementia after hip fracture surgery, with an odds ratio of 15.6 (95% CI 2.6–91.6) reported in one study [2, 3]. The pathophysiology of POD remains unclear, but a generally accepted model is of a combination of acute stressors (e.g. infection, inflammation, and hypoxaemia) superimposed on a 'vulnerable' brain [4]. We propose that the extent of acute stressors required to induce POD may be reduced in the presence of intrinsic brain vulnerability. A diagnosis of dementia (an example of intrinsic brain vulnerability) is associated with a relative risk of developing POD of approximately 2.8 [5]. However, the role of intrinsic brain vulnerability in the aetiology of POD in overtly cognitively intact people remains unclear.

Whilst the clinical features of neurodegeneration that occur in Alzheimer's disease and mild cognitive impairment can be screened for with appropriate bedside tests, some older people presenting for surgery may have neurodegeneration without overt clinical manifestations i.e. preclinical neurodegeneration. This is supported by magnetic resonance imaging (MRI) and positron electron tomography based research findings of structural changes or amyloid accumulation in the brain that may predate the clinical features of Alzheimer's dementia by a period up to a decade [6, 7].

We propose that some overtly cognitively intact older people who develop POD may already have preclinical neurodegenerative changes and the acute stressors associated with injury, surgery and anaesthesia serve to effectively unmask this, manifesting as POD. There are a small number of studies that have employed MRI to attempt identification of structural and functional anomalies in the brain that may predispose to POD. White matter hyperintensities and/or neurodegeneration have been identified in these studies as associations with POD [8]. The limited number of MRI studies, especially with regards to hip fracture surgery, may be

contributed to by the cost, limited access to, and practical challenges of, achieving brain MRI in people with hip fracture. A surrogate, which may aid screening for brain vulnerability and avoids the challenges of MRI, would provide increased opportunity to study the role of preclinical brain vulnerability in the aetiology of POD in cognitively intact older people.

Optical coherence tomography (OCT), a non-invasive scanning technique, can visualise retinal layers at near-microscopic level, permitting measurement of various retinal layers. Thinning of the circumpapillary retinal nerve fibre layer (cpRNFL) has been demonstrated in some neurodegenerative conditions, such as Alzheimer's disease, amnestic mild cognitive impairment and Parkinson's disease, albeit in varying extent [9–11]. Similarly, thinning of the retinal nerve fibre layer at the macular in people with Alzheimer's disease has also been noted by some investigators [12]. However, our review of the literature identified cpRNFL as the more frequently assessed in the context of mild cognitive impairment and Alzheimer's disease. Furthermore, thinning of the outer retinal layers has been associated with rare causes of dementia e.g. fronto-temporal dementia [13]. These conditions are all examples of intrinsic brain vulnerability. Given the interval between onset of neurodegeneration and the occurrence of clinical features, the notion that thinning of the retinal layers may be present in people with preclinical neurodegenerative changes is plausible. However, the integrity of the retinal layers can also be affected by a variety of ocular and systemic conditions such as age-related macular degeneration, glaucoma and diabetes; conditions that are common in the older age group [14]. The multitude of factors that may be associated with thinning of retinal layers may have implications for the utility of this modality as a screening tool for neurodegeneration, if efficacy is proven.

Standard OCT devices, while widely available in ophthalmology clinics and optometry services' are increasingly used, albeit mainly in research settings, in neurology. The standard device is table-mounted and requires a cooperative person be positioned sitting upright with supports for the chin and forehead. This position may be challenging for people with hip fracture. Recently, hand-held optical coherence tomography (HH-OCT) has become available to help clinicians obtain retinal images from participants who have difficulties accessing table-mounted OCT. HH-OCT is used in new-born and young children for research and to aid clinical diagnoses [15].

Given that OCT may detect retinal nerve fibre layer thinning in the setting of neurodegeneration, and neurodegeneration (overt or preclinical) may increase the risk of POD in older people, OCT may prove to be a useful adjunct to postoperative delirium risk assessment, especially if feasible at the patient's bedside.

With this study we assess the feasibility of bedside retinal imaging with HH-OCT, seek preliminary evidence for/against cpRNFL thickness as a marker of POD risk and assess utility if feasibility is demonstrated.

## Objectives

The primary objective was an assessment of the feasibility of acquiring retinal images using a HH-OCT at the patient's bedside in older people recovering from hip fracture surgery. The aim was an assessment of the ability to acquire retinal images without needing the participant to assume uncomfortable or challenging positions. The secondary objective was to determine if there is any difference in the cpRNFL thickness in people who have recovered from postoperative delirium compared to those who did not experience postoperative delirium. Finally, we assessed the proportion of our hip fracture population in whom there were exclusions (e.g. eye disease), to determine utility.

## Methods

### Study design

A prospective feasibility, observational cohort study involving people aged $\geq$ 65 years old recovering from primary hip fracture surgery in a secondary care centre. The study protocol received NHS HRA REC approval (reference 19/NW/0192). The report of the study's findings is guided by the Strengthening the Reporting of Observational Studies in Epidemiology (STROBE) guidelines.

### Setting

Participants were recruited from the trauma wards of Queen's Medical Centre, Nottingham. Batches of consecutive recruitment were used to accommodate the logistics of equipment availability and the challenges of the COVID pandemic. The periods of recruitment were Block 1: 30/08/2019–13/09/2019; Block 2: 03/02/2020–10/03/2020; Block 3: 08/08/2020–01/10/2020.

### Participants

During each period of recruitment, patients recovering from hip fracture surgery were consecutively screened for inclusion in this study. Inclusion criteria were age $\geq$ 65 years old, recovering from primary, unilateral hip fracture surgery, ability to comply and ability to give written informed consent. Exclusion criteria included known preoperative cognitive impairment (e.g. dementia, mild cognitive impairment, abbreviated mental test score (AMT) on admission $\leq$ 7), significant eye pathology (e.g. glaucoma, severe myopia i.e. < -6 diopters or severe difficulties in seeing far objects, diabetic retinopathy, retinal surgery, dense cataracts, age-related macular degeneration, eye trauma), postural or movement disorders precluding retinal imaging (e.g. severe tremor), intracranial pathology (head injury, previous cerebrovascular accidents or transient ischaemic attacks, multiple sclerosis, intracranial space occupying lesions, and neurodegenerative conditions such as Parkinson's disease), systemic malignancy, deteriorating clinical condition, multiple injuries, severe frailty, hip fracture sustained during hospital admission, and those not expected to survive the perioperative period. Recruitment occurred during the first week after surgery, after postoperative day two to allow recovery from the acute effects of the anaesthesia/surgical intervention (Fig 1).

### OCT image acquisition

For this study, we elected to acquire the retinal images after surgery. This was to reduce the burden on the participants, who are often in pain on admission and subsequently undergo interventions for pain management and preparation for urgent surgery. The cpRNFL thickness is stable in the short term, with an estimated age-related thinning of approximately 0.52μm/year [16]. This permitted the use of HH-OCT, in this study, after surgery and, where relevant, after an episode of POD had resolved.

Timing of OCT was dependent on participant convenience, ongoing clinical needs and occurrence of POD (deferred until after resolution). The retinal images were acquired using the HH-OCT device (Leica Microsystems, Envisu C2300, Wetzlar, Germany). Volumetric images were acquired using a 12 mm x 12 mm scan protocol (1000 A-scans x 100 B-scan). The main operator (AN, anaesthetics registrar) of the device was fully trained by the manufacturer and demonstrated the ability to attain high quality images prior to the study. When available, a second device operator (JS, technician) aided the main operator with image acquisition i.e. acquired the images while AN focussed the light beam. AN and JS had practised image

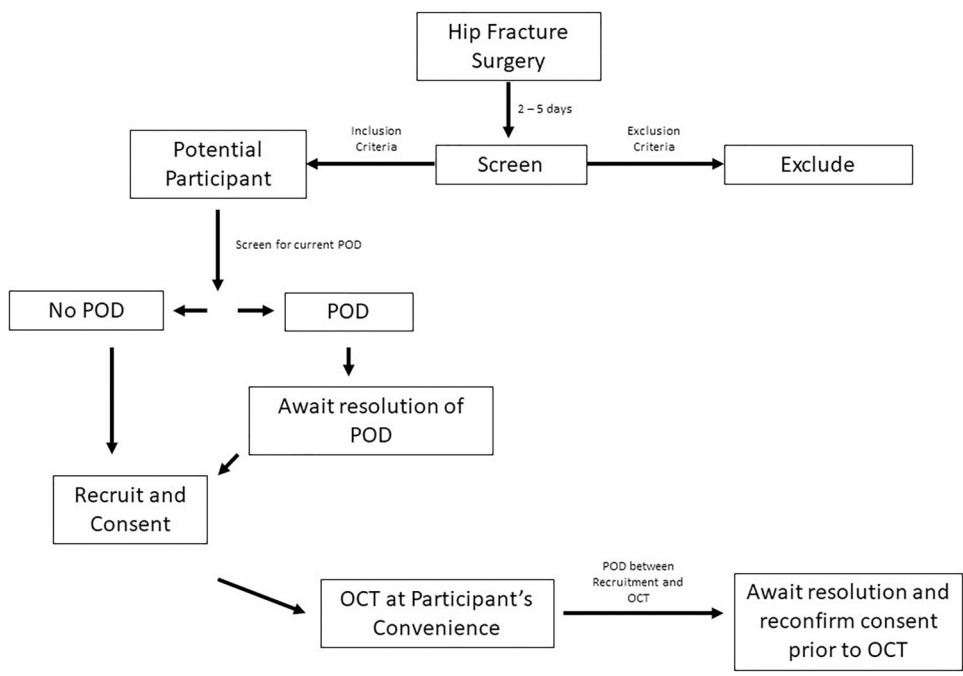

**Fig 1. Flowchart of participant recruitment.** POD = Postoperative Delirium, OCT = Optical Coherence Tomography.

acquisition on volunteers prior to the study. Participants were scanned at their bedside in the sitting or recumbent position based on participant convenience and preference. The right eye was the default eye for imaging acquisition and analysis. The left eye was used in participants in whom a good image could not be obtained from the right eye (e.g. dense cataract). Mydriatic agents were not used. Duration of image acquisition was recorded as were the number of repeats required to achieve a usable image. The images were segmented and analysed by AN and reviewed by ZT/FP whose expertise in OCT comes from a background of ophthalmic research using OCT and other methods [15].

**Image analysis.** ImageJ (http://imagej.nih.gov/ij/; National Institute of Health, Bethesda, Maryland) was used to batch convert the HH-OCT images into a file format that could be analysed using Copernicus SR software (Optopol Technology, Zawierci, Poland) [15].

Automated retinal segmentation was inspected and identified segmentation errors were manually corrected.

The cpRNFL thickness was measured at a 2.83mm diameter ring centred on the optic disc to adjust for the conversion of the image window. The superior, nasal, inferior and temporal cpRNFL quadrant thicknesses were also measured (Fig 2).

## Standard care

All participants were cared for on an acute trauma ward which operates a shared orthogeriatric care model, in line with national guidance. Surgery, anaesthesia and perioperative care follow national guidance, including senior surgical and anaesthetic care, prompt surgery, nerve block analgesia, formal assessment of delirium pre- and postoperatively and avoidance of deliriogenic drugs. In addition, assessment for behavioural changes that may suggest POD is routinely conducted by nursing staff at least twice daily.

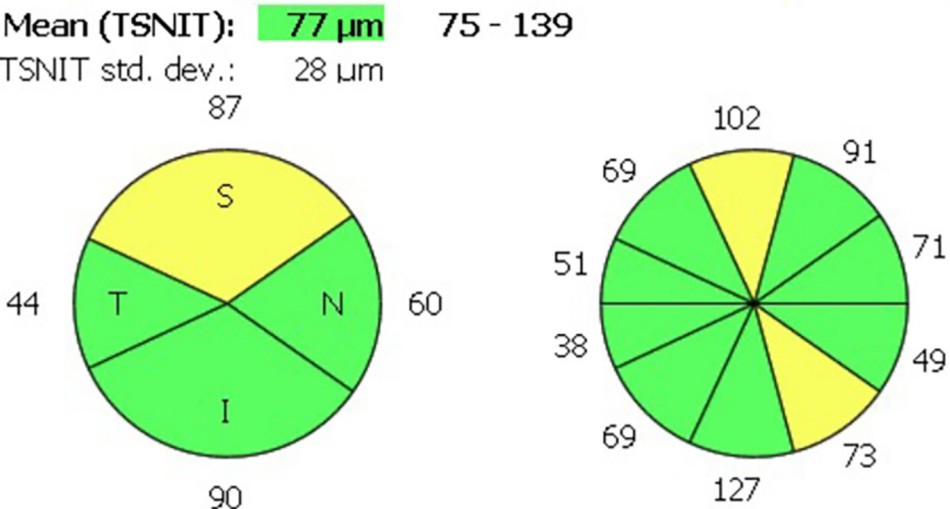

**Fig 2. HH-OCT circumpapillary retinal nerve fibre layer thickness measurement in a participant.** The cpRNFL thickness was divided into 10 radial segments (using the GDx Nerve Fiber Analyzer protocol, Carl Zeiss Meditec). The quadrants are created by amalgamating two (temporal), two (nasal), three (superior) and three (inferior) radial segments. The colours represent comparison to age-adjusted normative values (green = normal, yellow = borderline and red = out of range).

**Assessment of postoperative delirium.** After recruitment, participants' nursing and medical records were reviewed for evidence of prior POD, during the current admission. Thereafter POD assessments were conducted on a daily basis until HH-OCT was performed. A finding of POD by the researchers was communicated with the usual care team and resulted in postponement of HH-OCT till POD resolved. Assessment of POD status using DSM IV criteria was conducted by the researcher prior to HH-OCT. At discharge, participants' nursing and medical records were reviewed for evidence of POD after HH-OCT.

Participants were grouped into those without any evidence of POD and those who had POD based on review of the participant's medical and nursing records, researchers' assessment and discussion with the clinical team.

Retinal segmentation was performed without reference to delirium status and the OCT experts reviewing segmentation were blinded to delirium status.

## Outcomes

**Feasibility.** Primary feasibility: Handheld OCT image acquisition success rate was defined as the ability to achieve at least one retinal image from which the cpRNFL thickness could be measured.

**Secondary feasibility.** Number of attempts, time to acquisition and number of device operators.

**Exploratory outcome.** cpRNFL thickness (mean and quadrants) in no POD and POD groups.

**Utility outcome.** Proportion of patients recovering from hip fracture surgery during the study period that were excluded from participation due to pathology known to affect cpRNFL.

**Sample size.** As a feasibility study we chose a pragmatic sample size of 30 participants in line with guidance [17].

**Table 1. Recruitment pattern and demographics of participants and excluded people.**

|  | Whole cohort (n = 250) | Passed screening (n = 61) | Included (n = 30) |
|---|---|---|---|
| Sex (female / male) | 168/82 | 42/19 | 17/13 |
| Age in years: median(IQR) | 84 (78–89) | 81 (73–85) | 76 (70–81) |
| Experienced POD | Not assessed | Not assessed | 4 |

POD = Postoperative delirium

## Statistical methods

Feasibility and group proportions were reported using descriptive statistics. Between groups comparison of sex, AMTS and CCI were conducted using Chi-squared test. For analysis of cpRNFL thickness, the grouping variable was no POD vs POD In view of the low number of participants with POD in this study we have compared mean/quadrant cpRNFL thickness between groups using descriptive statistics (Mean and 95% confidence interval). We also reported the Mann-Whitney statistic.

## Results

There were 250 people of age $\geq$ 65 years recovering from primary hip fracture surgery at the research site of whom 61 fulfilled inclusion criteria. Of these, 21 patients declined, five were discharged prior to imaging, recruitment was halted by the COVID-19 pandemic in three, and two were transferred to another institution prior to scanning. This resulted in 30 participants in this study, 26 who had no POD and 4 who had POD (Table 1). The baseline characteristics of included participants are depicted in Table 2.

### Feasibility

Primary feasibility: Of 30 participants in this study, retinal images from which all segments of cpRNFL thickness could be analysed were achieved in 25 participants (83%). An additional participant had clear images, but the inferior part of the measurement area was obscured. This final participant's data were included in analysis of retinal segments with available data. The retinal images were too dark to segment in four participants and were excluded from the exploratory analysis.

**Table 2. Baseline characteristics of participants.**

|  | No POD | POD | p-value |
|---|---|---|---|
| Female/Male | 14/12 | 3/1 | 0.427 |
| Age in years +/- SD | 76 +/- 8 | 81+/- 4 | 0.2 |
| AMTS score(n) | 8(1)<br>9(5)<br>10(20) | 8(1)<br>9(1)<br>10(2) | 0.256 |
| CCI grade (n) | Mild (2)<br>Moderate (13)<br>Severe (11) | Mild (0)<br>Moderate (2)<br>Severe (2) | 0.837 |

AMTS = Abbreviated mental test score, POD = postoperative delirium, CCI = Charlson Comorbidity Index
CCI Grade = Mild (CCI Score 0–2), Moderate (CCI score 3–4), Severe (CCI score $\geq$ 5)

**Table 3. Exploratory analysis of mean and segmental circumpapillary retinal nerve fibre thickness between participants who did not have postoperative delirium and those who had postoperative delirium (Descriptive statistics and Mann-Whitney U statistic).**

| | No POD (n = 22)* Mean (95% CI) | POD (n = 4) Mean (95% CI) | Mann-Whitney U statistic (p-value) |
|---|---|---|---|
| Mean cpRNFL μm | 89.19 (85.41–92.97) | 76.50 (62.60–90.40) | 9.0 (0.011) |
| Superior cpRNFL μm | 102.24 (97.92–106.56) | 85.50 (70.89–100.11) | 7.0 (0.005) |
| Nasal cpRNFL μm | 67.67 (61.75–73.58) | 56.25 (43.83–68.67) | 23.0 (0.150) |
| Inferior cpRNFL μm | 103.67 (97.00–110.33) | 91.75 (70.87–112.63) | 25.0 (0.231) |
| Temporal cpRNFL μm | 53.86 (51.26–56.45) | 46.25 (35.83–56.67) | 14.0 (0.032) |

*N = 22 except mean cpRNFL and inferior cpRNFL where n = 21

POD = postoperative delirium, cpRNFL = circumpapillary retinal nerve fibre layer, CI = confidence interval.

Secondary Feasibility: Participants' positions during scanning were lying in bed 23 (76.7%) participants, sitting on a chair 6 (20%) participants and sitting on edge of bed 1(3.3%) participant.

Based on availability, there were two scanner operators for 21 participants and one for the remaining nine participants. The four participants excluded from the exploratory analysis due to dark images had two scanner operators.

While the default eye was the right eye, the image used for analysis was the left eye in four participants (unclear right eye images x 2, right eye surgery, and right eye dense cataract).

The number of attempts (median (IQR)) to achieve a scan for analysis was 2 (2–4).

Time to achieve scan for analysis was available in 24 of 25 participants with fully analysable scans. The median (IQR) time was 3 (2–4.25) minutes.

Exploratory outcome: Circumpapillary Retinal Nerve Fibre Layer thickness (cpRNFL) measurements

Mean and inferior cpRNFL was measured in 25 participants of whom 4 had postoperative delirium. Superior, nasal and temporal segment data was available in 26 participants. Analysis was performed at a circumpapillary ring with an inner ring diameter of 2.83mm and a ring thickness of 0.10mm.

The findings are depicted in Table 3, and Figs 3, 4

## Utility outcome

Of 250 people recovering from primary hip fracture surgery during the study period, 189 had comorbidities that precluded participation in this study. Although the majority of excluded patients had multiple exclusion criteria we report here on the most significant exclusion criterion for each participant. The primary reason for exclusion are outlined in Table 4.

## Discussion

Acquisition of retinal images from which cpRNFL thickness can be measured, using a hand-held OCT device, at the patients' bedside is feasible. The exploratory analysis suggests that in a carefully selected cohort, the average retinal nerve fibre layer thickness in participants who had postoperative delirium may be thinner than in participants who did not have postoperative delirium; however the study was not powered for this outcome. That said, approximately half the patients with primary hip fracture admitted to our institution during the study period had other potential causes of retinal nerve fibre layer thinning; precluding the use of OCT in the

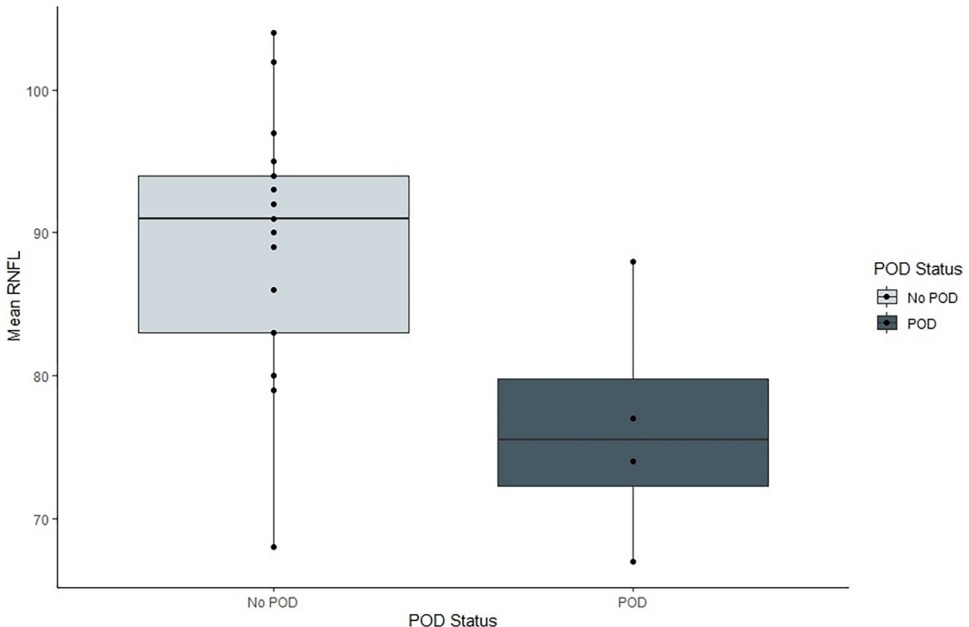

**Fig 3. Box plot of Mean cpRNFL (µm) at Ring diameter 2.83mm in participants who did not have postoperative delirium and participants who had postoperative delirium with individual participant's values.**
POD = postoperative delirium, RNFL = retinal nerve fibre layer.

context of this study. In these the alternative brain vulnerability imaging test remains brain CT or MRI scan.

Reduction of POD risk in older people is an urgent problem given the associated adverse outcomes. Multicomponent, non-pharmacological interventions have been shown to reduce

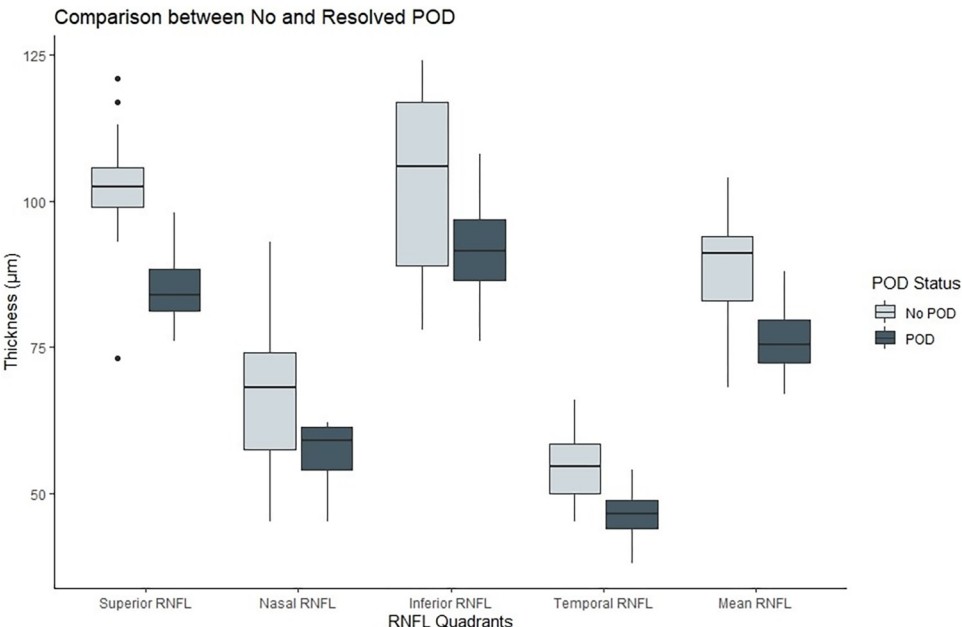

**Fig 4. Boxplot of cpRNFL thickness (µm) by Quadrants at ring diameter 2.83mm comparing participants who had postoperative delirium to those who did not experience postoperative delirium.** POD = Postoperative delirium, RNFL = Retinal Nerve Fibre Layer.

**Table 4. Reasons for exclusion.**

| Primary reason for exclusion | Number |
|---|---|
| Inability to give informed consent (dementia, cognitive impairment or prevalent delirium) | 89 |
| Local eye pathology (blind, glaucoma, macular degeneration, retinal surgery or severe cataracts) | 21 |
| Systemic/neurodegenerative disease with known associations with retinal pathology (Diabetes, Parkinson's disease) | 20 |
| Brain pathology (head injury, tumours, CVA, recurrent TIA) | 21 |
| Acute clinical status (current cancer, clinical deterioration) | 29 |
| Inability to comply (frailty, multiple injuries, severe tremor) | 5 |
| Hip fracture sustained during hospital admission | 4 |

CVA = Cardiovascular Accident, TIA = transient Ischaemic Attack

129 patients (52%) were excluded due to presence of local, neurodegenerative or systemic conditions that may independently affect the retinal nerve fibre layer thickness.

the incidence of delirium and POD [18]. However the resource burden of application of these interventions may prove challenging if used in a non-targeted manner. Identification of high risk patients may aid targeted application of these proven interventions, thus reducing both the resource burden of the intervention and the POD risk of the most vulnerable. This mandates accurate POD risk prediction tools. While there are a number of POD prediction tools, there is room for improvement in the predictive capability of these tools [19, 20]. Overt features of neurodegeneration such as cognitive impairment and Folstein's mini mental state examination are included in some tools, but there is an absence of object measures of covert, preclinical neurodegeneration. This is not surprising given the expense of screening cognitively intact older people scheduled for surgery with brain MRI/CT scans. The development of an inexpensive bedside tool that could act as a surrogate for screening for preoperative covert neurodegeneration may further improve POD risk assessment.

With this study we have taken the first step i.e. feasibility, of exploring the potential of HH-OCT as a bedside tool that may be used to aid POD risk prediction. The result of our exploration of differences in cpRNFL thickness between POD and no POD suggests this warrants further investigation. It must however be recognised that utility may be limited to those people who do not have comorbidities that influence cpRNFL.

Whilst feasibility has been shown, this study has some limitations. The HH-OCT device does not have an integrated automated retinal segmentation software. Images therefore needed to be exported and formatted for a non-platform segmentation software. This limited the ability to predict, during acquisition of images at the participant's bedside, the final image clarity, after conversion for the non-platform segmentation software. As a result four participants' retinal images were excluded from analysis as they were too dark for segmentation by the software. Also, the acquisition of clear images is enhanced by calibrating the device with each participant's refractive index where applicable. Unfortunately, our glasses-wearing participants did not have this information readily available. Nevertheless, the main purpose of the study was to demonstrate feasibility, not to provide a definitive assessment of between group differences.

Our experience is that retinal imaging using HH-OCT at the patient's bedside is easy to learn and quick to implement.

## Conclusion

In conclusion, retinal imaging at the patient's bedside, using the hand-held OCT is feasible. Our data suggest that in the absence of comorbidities that alter the cpRNFL (e.g. eye disease),

thinning of the cpRNFL may be associated with an increased risk of postoperative delirium. This needs to be tested in future prospective studies.

## Supporting information

**S1 File.**
(XLSX)

## Acknowledgments

Thank you to the trauma wards staff at Queen's Medical Centre.

## Author Contributions

**Conceptualization:** Abiodun M. Noah, Cris S. Constantinescu, Irene Gottlob, Dorothee P. Auer, Rob A. Dineen, Iain K. Moppett.

**Data curation:** Abiodun M. Noah, Frank Proudlock.

**Formal analysis:** Abiodun M. Noah, Zhanhan Tu, Frank Proudlock.

**Funding acquisition:** Cris S. Constantinescu, Dorothee P. Auer, Iain K. Moppett.

**Investigation:** Abiodun M. Noah, Jennie Spendlove.

**Methodology:** Abiodun M. Noah.

**Supervision:** Zhanhan Tu, Iain K. Moppett.

**Writing – original draft:** Abiodun M. Noah.

**Writing – review & editing:** Jennie Spendlove, Cris S. Constantinescu, Irene Gottlob, Dorothee P. Auer, Rob A. Dineen, Iain K. Moppett.

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
