## [Decision Letter · Decision Letter 0]

22 Apr 2024

PONE-D-24-05686Retinal imaging with hand-held optical coherence tomography in older people with or without postoperative delirium after hip fracture surgery: A feasibility studyPLOS ONE

Dear Dr. Noah,

Thank you for submitting your manuscript to PLOS ONE. After careful consideration, we feel that it has merit but does not fully meet PLOS ONE’s publication criteria as it currently stands. Therefore, we invite you to submit a revised version of the manuscript that addresses the points raised during the review process. I apologize that this review has taken so long to return.  I was unable to obtain a second review despite many inquiries, but I believe you will find the comments of the single reviewer helpful in revising your paper. Please submit your revised manuscript by Jun 06 2024 11:59PM. If you will need more time than this to complete your revisions, please reply to this message or contact the journal office at plosone@plos.org. Please include the following items when submitting your revised manuscript:A rebuttal letter that responds to each point raised by the academic editor and reviewer(s). You should upload this letter as a separate file labeled 'Response to Reviewers'.A marked-up copy of your manuscript that highlights changes made to the original version. You should upload this as a separate file labeled 'Revised Manuscript with Track Changes'.An unmarked version of your revised paper without tracked changes. You should upload this as a separate file labeled 'Manuscript'.If applicable, we recommend that you deposit your laboratory protocols in protocols.io to enhance the reproducibility of your results. Protocols.io assigns your protocol its own identifier (DOI) so that it can be cited independently in the future. For instructions see: https://journals.plos.org/plosone/s/submission-guidelines#loc-laboratory-protocols. Additionally, PLOS ONE offers an option for publishing peer-reviewed Lab Protocol articles, which describe protocols hosted on protocols.io. Read more information on sharing protocols at https://plos.org/protocols?utm_medium=editorial-email&utm_source=authorletters&utm_campaign=protocols.

We look forward to receiving your revised manuscript.

Kind regards,

Alfred S Lewin, Ph.D.

Section Editor

PLOS ONE

Journal Requirements:

"Dr Noah’s PhD – British Journal of Anaesthesia/Royal College of Anaesthetists project grant. Grant holder - Prof Iain Moppett

Purchase of OCT – BJA/RCOA grant, Precision Imaging Beacon Centre and Academic Department of Neurology, University of Nottingham."

4. In the online submission form, you indicated that "The data underlying the results represented in the study are available from Dr Abi Noah"

**Additional Editor Comments:**

Please address the reviewers concerns about the limitations of obtaining OCT images by a single operator.

Reviewers' comments:

Reviewer's Responses to Questions

**Comments to the Author**

1. Is the manuscript technically sound, and do the data support the conclusions?

Reviewer #1: Yes

2. Has the statistical analysis been performed appropriately and rigorously? 

Reviewer #1: Yes

3. Have the authors made all data underlying the findings in their manuscript fully available?

Reviewer #1: Yes

4. Is the manuscript presented in an intelligible fashion and written in standard English?

Reviewer #1: Yes

5. Review Comments to the Author

Reviewer #1: Revision

Retinal imaging with hand-held optical coherence tomography in older people with or

without postoperative delirium after hip fracture surgery: A feasibility study

General comments:

The authors present a feasibility study on the assessment of the peripapillary nerve fiber layer (RNFL) thickness in patients experience postoperative delirium (DOL) after surgery for unilateral hip fracture. Despite a very small sample of 26 patients without and 4 patients with DOL the authors find significant differences in RNFL thickness between the two groups. While this certainly needs verification in a much larger sample, these results are very much of interest.

Major comments:

The images were taken by one single operator who was in parts assisted by a colleague. Acquisition of OCT images using a handheld device can be very challenging for various reasons. It would have been very insightful if two people had been acquiring the images and their agreement was assessed. Same counts for retinal layer segmentation, especially if a custom tool for ImageJ was used to the assessments.

Minor comments:

p.3 l.55:

Define “older”.

P.3. L.60:

Add a reference for POD hypothesis.

P.4. l.88:

Thinning of the inner retinal layers (most notably RNFL and the IPL-GCL complex) has also been shown on macular OCT. Please include this and explain why it was chosen to assess the peripapillary RNFL. OCT acquisition of the macula is generally easier due to fixation.

P. l.88:

There is evidence of a thinning of the outer retinal layers in frontotemporal dementia. This should be included in the introduction.

P.5 l.93:

Add references.

P.6 l.138:

Define “severe myopia”. Generally, <-6 diopters is used.

P.8. L. 186:

I assume the age-adjusted normative values were taken from a measurements using a different OCT devise. This should be acknowledged and included in the limitations.

P.9. L. 206:

Please define who the “OCT experts” were and how they were trained.

Table 2:

Please include lens status in the table.

p.16 l.312:

Please stick to “cpRNFL” as the abbreviation.

Figure 2:

The size of the segments varies. Please either unitize or explain in detail why this is the case.

6. PLOS authors have the option to publish the peer review history of their article (what does this mean?). If published, this will include your full peer review and any attached files.

Reviewer #1: **Yes: **Lukas Goerdt

---

## [Author Response · Author response to Decision Letter 0]

6 Jun 2024

Thank you for your review of our work

This is our response to the reviewer

Your comment

General comments:

The authors present a feasibility study on the assessment of the peripapillary nerve fiber layer (RNFL) thickness in patients experience postoperative delirium (DOL) after surgery for unilateral hip fracture. Despite a very small sample of 26 patients without and 4 patients with DOL the authors find significant differences in RNFL thickness between the two groups. While this certainly needs verification in a much larger sample, these results are very much of interest.

Our response

Thank you. We agree that a larger study is required to confirm or refute our findings

Major comments

The images were taken by one single operator who was in parts assisted by a colleague. Acquisition of OCT images using a handheld device can be very challenging for various reasons. It would have been very insightful if two people had been acquiring the images and their agreement was assessed. Same counts for retinal layer segmentation, especially if a custom tool for ImageJ was used to the assessments.

Our response

Indeed, it was challenging but actually quick to get used to

My training with the handheld device by the Leica engineer and also by our regional researchers who use handheld OCT was as a single user. Hence my approach of single operator albeit assisted in parts by a second operator.

Agreed, it would have been useful to have two people independently acquire images and compare their agreement. Unfortunately we didn’t. Perhaps in a future study. 

Minor comments:

p.3 l.55:

Define “older”.

Our response

Thank you added

Minor comment

P.3. L.60:

Add a reference for POD hypothesis.

Our response

Thank you added

Your comment

P.4. l.88:

Thinning of the inner retinal layers (most notably RNFL and the IPL-GCL complex) has also been shown on macular OCT. Please include this and explain why it was chosen to assess the peripapillary RNFL. OCT acquisition of the macula is generally easier due to fixation.

Our response

Thank you.

Prior to this study we conducted a systematic review on OCT and MCI [Ref Noah]. The majority of studies our literature review identified were circumpapillary. Given that we are looking for potential preclinical changes that may evolve to cognitive impairment, we elected to remain circumpapillary in this study. 

Your comment

P. l.88:

There is evidence of a thinning of the outer retinal layers in frontotemporal dementia. This should be included in the introduction.

Our response

Thank you we have added this. 

Your comment

P.5 l.93:

Add references.

Our response

Thank you we have added this

Your comment

P.6 l.138:

Define “severe myopia”. Generally, <-6 diopters is used.

Our response

Thank you.

WE have clarified this in the manuscript in the manuscript

Your comment

P.8. L. 186:

I assume the age-adjusted normative values were taken from a measurements using a different OCT devise. This should be acknowledged and included in the limitations.

Our response

Thank you. 

Your comments

P.9. L. 206:

Please define who the “OCT experts” were and how they were trained.

Our response

These are Ophthalmologists who conduct research using the handheld OCT device. They work out of the Ulverscroft eye Unit, University of Leicester and are co-authors in this work. 

Comment

Table 2:

Please include lens status in the table.

Our response

Thank you for your comment. We acknowledge that some of our participants wear glasses. Unfortunately they could not remember what their refractive index was so we did not collect this data. This has been acknowledged as one of the limitations of the work.

Your comment

p.16 l.312:

Please stick to “cpRNFL” as the abbreviation.

Our response

Thank you. 

Your comment

Figure 2:

The size of the segments varies. Please either unitize or explain in detail why this is the case.

Our response

Thank you

This is consistent across all the images we obtained from the software

A quick search for Copernicus OCT also shows images with this ‘asymmetry’ 

Thank you

---

## [Editor Report · Decision Letter 1]

9 Jun 2024

Retinal imaging with hand-held optical coherence tomography in older people with or without postoperative delirium after hip fracture surgery: A feasibility study

PONE-D-24-05686R1

Dear Dr. Noah,

We’re pleased to inform you that your manuscript has been judged scientifically suitable for publication and will be formally accepted for publication once it meets all outstanding technical requirements.

Kind regards,

Alfred S Lewin, Ph.D.

Section Editor

PLOS ONE
---

## [Editor Report · Acceptance letter]

8 Jul 2024

PONE-D-24-05686R1 

PLOS ONE

Dear Dr. Noah, 

I'm pleased to inform you that your manuscript has been deemed suitable for publication in PLOS ONE. Congratulations! Your manuscript is now being handed over to our production team.

Kind regards, 

on behalf of

Dr. Alfred S Lewin 

Section Editor

PLOS ONE